# Therapeutic Implications of Probiotics in the Gut Microbe-Modulated Neuroinflammation and Progression of Alzheimer’s Disease

**DOI:** 10.3390/life13071466

**Published:** 2023-06-28

**Authors:** Toshiyuki Murai, Satoru Matsuda

**Affiliations:** 1Graduate School of Medicine, Osaka University, 2-2 Yamada-oka, Suita 565-0871, Japan; 2Department of Food Science and Nutrition, Nara Women’s University, Kita-Uoya Nishimachi, Nara 630-8506, Japan

**Keywords:** gut microbiome, fatty acid, probiotics, amyloid, tauopathy, antibiotics

## Abstract

Alzheimer’s disease (AD) is characterized by the accumulation of specific proteins in the brain. A recent study revealed that manipulating gut microbiota (GM) significantly reduced tau pathology and neurodegeneration in an apolipoprotein E isoform-dependent manner. The resilience of a healthy microbiota protects it from a variety of dysbiosis-related pathologies. Convincing evidence has demonstrated the roles of GM in the pathogenesis of AD, which are partly mediated by modified microglial activity in the brain. Therefore, modulation of GM may be a promising therapeutic option for AD prevention. In addition to providing the cells with energy and affecting microglial maturation, these microbial metabolites appear to influence neuronal function. One of the potential therapeutic approaches targeting GM may involve using probiotics. Additionally, human GM and its metabolites have also become potential therapeutic targets for developing interventions for the prevention of disorders. Synbiotics and postbiotics can also be used to treat AD by modulating GM. In addition, physical activity, exercise, and physical fitness are being considered as potential nonpharmacological therapies to reduce signaling pathways related to neuroinflammation. Therefore, interventions targeting GM might be promising strategies for health promotion.

## 1. Introduction

Neuroinflammation, such as microgliosis and astrogliosis, is one of the major hallmarks and pathological features associated with a wide variety of neurodegenerative disorders, including Alzheimer’s, Parkinson’s, and Huntington’s diseases [1]. The role of neuroinflammation in these neurodegenerative diseases is definitively demonstrated in prototypical neuroinflammatory diseases, such as multiple sclerosis, in addition to the invariable occurrence of inflammation in the brain. Among them, Alzheimer’s disease (AD) is the most common form of dementia, whose typical diagnostic feature is the increased deposition of neurofibrillary plaques in the brain. These insoluble protein aggregates mainly consisting of amyloid-β (Aβ) protein with an abnormal structure and hyper-phosphorylated tau proteins [1]. A pharmacological therapy effective in the treatment of AD by impeding the progression of the disease has not yet been established, and the only options available are restricted to symptomatic interventions that slow down the progression of the disease [1]. Thus, the development of novel therapies effective in mitigating AD and the elucidation of their neuroprotective mechanisms are intriguing for researchers. Although the stage and the mechanisms by which neuroinflammation arises in the course of the progression of AD have not yet been fully resolved, numerous efforts have been made to elucidate the pathological and physiological role of inflammation in this disease by taking into account the multiple interactions of inflammatory mediators [2].

## 2. Oxidative Stress and Protein Aggregation in AD

One of the mechanisms confirmed in an experimental model of AD involves the reduction in oxidative damage-induced reactive oxygen species (ROS) [3]. ROS, as the name suggests, is a family of reactive species derived from molecular oxygen that are both continuously produced and scavenged in the cells of all aerobic organisms. Superoxide anion, hydroxyl radicals, alkoxyl radicals, and peroxyl radicals are the major free radical-type ROS, while hydrogen peroxide, ozone, singlet molecular oxygen, electronically excited carbonyls, and organic hydroperoxide are the major non-free radical forms of ROS. The excessive generation of ROS plays a critical role in the pathogenesis of diseases, and the improper regulation of ROS levels contributes to the pathologies of inflammation, cancer, and neurodegeneration. ROS are harmful to cell components and cause DNA damage [4]. Oxidative damage in cells is mainly attributed to the excess production of ROS. ROS produce metabolic intermediates that are involved in various signaling pathways. It has been reported that abnormal Aβ inhibited the long-term potentiation (LTP) in neurons that were saved with the administration of an antioxidant, suggesting a synergistic achievement between the progression of the pathogenesis of AD and oxidative stress [1]. Oxidative stress occurs and is a constant feature of AD brain pathology. Additionally, oxidative stress has a contributing role to neuroinflammation and the pathogenesis of AD [5]. ROS also act as secondary messengers in the transmission of redox-sensitive signals, including the stress-activated mitogen-activated protein kinases (MAPKs), p38 MAPKs, and c-Jun N-terminal kinases (JNKs). Thus, ROS has a dual function, with the other being to promote the activity of Akt. The Akt pathway can be induced through inhibition of the counteracting phosphatase and tensin homolog (PTEN). Events triggered via the phosphatidylinositol-3 kinase (PI3K) pathway involve the activation of nicotinamide adenine dinucleotide phosphate oxidase (NADPH) and the production of ROS. Thus, ROS are central to cellular redox regulation and exert positive feedback on the PI3K signaling pathway through mechanisms including the reversible oxidation and inactivation of PTEN and other phosphatases that negatively regulate this pathway.

Studies have shown that dietary choices could play a role in protecting against the neuroinflammation associated with AD [3]. However, the exact relationship between the consumption of nutrients and its neuroprotection effects remains largely unknown. In addition, it is difficult to examine the distinct effects when diet is involved. Although a variety of lifestyle factors may affect brain function, the regulation of food-related facets might be a promising strategy for preventing brain dysfunction. For example, there are reports showing that the consumption of coffee seems to ameliorate AD-induced cognitive impairment and decrease Aβ levels [6]. Thus, nutraceuticals with antioxidant activity, such as phenolic compounds present in coffee beans, may have beneficial effects in the prevention of AD.

AD is characterized by the accumulation of specific proteins in the brain. Aβ protein is accumulated in the extracellular space, and tau protein is accumulated as intracellular aggregates [6]. The only established biomarkers are Aβ (1–42), total tau, and phosphorylated tau in cerebrospinal fluid. The amyloid hypothesis proposes amyloid-β protein accumulation as the main cause of the disease. This protein is the main component of plaques and is derived from a longer type I membrane glycoprotein, amyloid precursor protein (APP). Aβ is a protein with 39 to 43 amino acid residues long originating from the C-terminal region of the APP by proteolytic processing. The cleavage of the APP by α-secretase releases a soluble APP-α from the cell surface and leaves an 83-amino-acid-long C-terminal APP fragment. The amyloidogenic processing of APP is executed through sequential cleavages by β- and γ-secretases at the N- and C-termini of Aβ, respectively. APP is produced in most peripheral cells, and Aβ is present in blood plasma in addition to the cerebrospinal fluid. There are two main isoforms of Aβ peptide in humans: Aβ (1–40) and Aβ (1–42); the former is more abundant, but the latter is able to form fibrils more rapidly and is considerably more neurotoxic. Aβ is predominantly expressed at the plasma membrane and transported to the extracellular space, where Aβ is deposited as protein deposits, called senile plaques, which are a characteristic feature of AD. The toxicity of Aβ is attributed to fibrillar Aβ, which is prone to damage neurons or initiate an intracellular signaling cascade toward neuronal cell death. Both isoforms are known to inhibit long-term potentiation (LTP) in neurons and support an enhancement of synaptic efficacy after brain high-frequency stimulation (HFS). Tau proteins are microtubule-associated proteins that are predominantly expressed in neurons. Tau-positive neurofibrillary lesions constitute mainly a neuropathological feature of AD. Although these proteins are considered hallmarks of the disease, brain atrophy only correlates highly with tau protein accumulation and not the deposition of Aβ protein [7]. The complex nature of the central nervous system (CNS) requires certain specialized immunological adaptations to detect and respond to environmental changes, and the local microenvironment in the brain related to tauopathy is often instructive for the recruitment and guidance of the transformation of T cells [8].

## 3. Gut Microbiota (GM) and Tauopathy

The maintenance of healthy GM is one of the important factors for the maintenance of the immune system and cognitive–emotional balance via the production of many biologically active metabolites, giving rise to the GM–brain axis [9]. The composition of GM generally exhibits a high variation among individuals, and once the diversity of the commensal microbiota is established to some extent during childhood, it exhibits strong resilience, i.e., its composition and activity subsequently remain substantially stable. This resilience of the health-promoting microbiota protects the host from a variety of pathologies related to dysbiosis [9]. Thus, interventions targeting them might be promising strategies for treating diseases and promoting health.

GM is reported to be closely related to a variety of cancers and neurological disorders. Intestinal dysbiosis also favors the growth of certain species of gut bacteria, which increases the risk of certain types of cancer through numerous mechanisms, including the production of many kinds of biological factors that degrade the products of tumor suppressor genes, the generation of oxidative stress, the activation of proinflammatory mechanisms, the alteration of cell proliferation, the modulation of survival pathways, and the alteration of the immune system [10]. Because of its influence on disease development and prognosis, the microbiota has become a target in the field of therapy [11]. Currently, the available evidence supports this association, and recent clinical studies regarding the use of antimicrobial agents in patients with AD have also been summarized [12].

The most well-known biological marker of neurodegeneration is the accumulation of misfolded and aggregated proteins in the brain [13]. These aggregates are often surrounded by certain immune cells, including microglia and astrocytes [14]. Patients with neurodegenerative disorders such as AD, Parkinson’s disease, and amyotrophic lateral sclerosis commonly exhibit alterations in their immune systems and the profile of the bacterial communities that inhabit their guts. GM can regulate gene expression in microglia in a manner correlated with the status of apolipoprotein E [14]. The apolipoprotein E ε4 allele mapped to chromosome 19q13.2 is the first gene to be identified and associated with a significantly enhanced risk of sporadic late-onset AD [15]. However, it is not clear whether the disruption of GM is a cause or a result of neurodegeneration or whether the timely treatment of this gut dysbiosis could impede its progress [15]. The most widely accepted model explaining the progress of AD suggests that Aβ pathology may be an upstream event in AD and function as a trigger of downstream pathways, including tau-mediated toxicity, the misfolding of phosphorylated tau proteins, their accumulation in tangles, and the proliferation of tau proteins that leads to cortical neurodegeneration [16].

A recent study revealed that the manipulation of GM resulted in significant mitigation of tau-related pathology and neurodegeneration in an apolipoprotein E isoform-dependent manner. Holtzman and colleagues at Washington University found that disruption of GM affects neuronal loss in a mouse tauopathy model and that this effect is dependent on the expression of apolipoprotein E4 [17]. Astrocyte and microglial morphology and their transcriptomic analyses revealed that the manipulation of GM drives glial cells to a more homeostatic-like state, which indicates that GM strongly influences neuroinflammation and tau-mediated neurodegeneration [17]. Further, microbiome and metabolite analyses suggested that microbially produced short-chain fatty acids (SCFAs) are mediators of the neuroinflammation–neurodegeneration axis and their supplementation resulted in an enhanced reactive microglial morphology and an increase in tau pathology [17]. Exclusively, the males of all apolipoprotein E genotypes that were treated with antibiotics had reduced phosphorylated tau levels, a significant attenuation of atrophy, and a marked improvement in nest-building behavior, albeit with an enhanced effect in mice expressing apolipoprotein E3 rather than E4. These results are reminiscent of previous findings, which report that long-term antibiotic treatment leads to a reduction in Aβ pathology in only male mice in an amyloid model [17].

The role of GM in the pathogenesis of AD, which is partially mediated by altered microglial activity in the brain, has been demonstrated by compelling evidence [18]. In fact, microglial dysfunction has been detected in a variety of neurodegenerative disorders, including AD. The GM–glia–brain-immune axis might be influenced by the production of inflammatory cytokines and/or the reduction in the levels of certain metabolite compounds, such as SCFAs, thus modulating the regulation of the sympathetic afferent nerve and glial cells [18]. For example, butyric acid, being a key SCFA, might be associated with a favorable response in the treatment of schizophrenia, suggesting a pivotal role in the GM–brain axis [18]. Prebiotics are specific types of plant fibers that may stimulate the growth of healthy bacteria in the gut, while probiotics usually contain specific species of microorganisms that directly promote the growth of health-beneficial microbes in the gut [18]. Furthermore, mild physical exercise has a positive effect on GM with a wider diversity, which may also improve the symptoms of major depression. Metabolites derived from microbial fermentation could mediate their effects via immunological and neuroendocrine mechanisms [18]. In particular, microbial fatty acid metabolites, such as SCFAs, could contribute to the synergism between the consortium of intestinal microbes and systemic immune cells [18], probably in part through epigenetic mechanisms. Notably, GM–brain communication is considered to be bidirectional. In addition, supplementation of probiotic materials has been shown to improve the cognition of recipients with AD [18]. It has been revealed that probiotic supplementation, including *Bifidobacterium bifidum* and *B. longum*, in patients with AD could improve their cognitive function [18]. Therefore, the modulation of GM might be a promising therapeutic option to prevent AD [18].

GM could play a crucial role in promoting human health and ameliorating various diseases [19]. While *Bacteroides* sp. and *Pseudomonas* sp. can induce colitis, certain bacterial species may enhance the development of a good GM that helps in the inhibition of carcinogenesis [19]. We obtained a preliminary result implying that a specific type of probiotics can exhibit suppressive effects against inflammatory bowel disease-related carcinogenesis [19]. This might provide definitive evidence that probiotics consisting of specific strains of bacteria could prevent the development and inhibit the progression of inflammatory bowel disease-related tumors [19]. In addition, probiotics with *Clostridium butyricum* have promoted the proliferation of macrophages that produce interleukin-10, which could promote the inhibition of inflammation in mouse intestines [20].

Microbe-derived metabolites not only provide cells with a source of energy and affect microglial maturation, they might also have the ability to influence neuronal function. SCFAs may modulate the levels of secretory neurotransmitter molecules and neurotrophic factors. For instance, acetate has previously been shown to alter the levels of neurotransmitter release including glutamate, glutamine, and γ-amino butyric acid (GABA) in the hypothalamus and increase neuropeptide expression [18]. Propionate and butyrate, the major SCFAs other than butylate, exert a distinct influence on the intracellular potassium ion level, implying the involvement of SCFAs in the modulation of cell signaling systems [18]. Time-dependent eating restrictions, which limit the daily timing for meals to a window of 6–12 h, have been shown to reduce the risks of cardiometabolic diseases through the regulation of circadian rhythms in metabolic and physiological pathways [18].

## 4. Identification of Possible Probiotics for the Treatment of AD

The potential approaches to practical therapeutic intervention for the improvement of microbiota-related diseases might target GM and/or their metabolites by following the alteration of pathological pathways in diseases [19]. One of these approaches may involve the utilization of pathways in the progression of diseases [19], while another may involve the utilization of probiotics (Table 1). A meta-analysis indicated that probiotics consumption enhanced cognition in subjects with AD, possibly by decreasing the levels of inflammatory and oxidative biomarkers [21]. Another meta-analysis showed that intestinal microbiota balance therapy supported by probiotics could improve the cognitive function of patients with Alzheimer’s disease [22].

Human GM and its metabolites also serve as potential targets for the development of therapeutic interventions for the prevention of such disorders. SCFAs are carboxylic acids containing fewer than six C atoms, such as acetic acid (CH_3_COOH), propionic acid (CH_3_CH_2_COOH), and butyric acid (CH_3_CH_2_ CH_2_COOH), are produced by GM, mainly through fermentation [20]. For example, a probiotic formulation of lactic acid bacteria and bifidobacteria (SLAB51) administration exerts multiple effects by modulating gut microbiota composition and causing metabolic changes, such as the increase in SCFAs, which are able to directly act in the gut and the brain due to their ability to pass the blood–brain barrier [22]. Recently, engineered probiotics have demonstrated their potential applicability as a novel type of drug delivery system that could effectively prevent inflammatory diseases [19]. These results further support the rational manufacturing of new types of probiotics for the targeted treatment of disorders [19]. Particularly, the anti-inflammatory molecules identified in preclinical and clinical studies may be of pivotal importance in providing insights into the identification of novel therapeutic targets for the practical application of genetically engineered probiotics [19]. In addition, engineered probiotics could serve as optimal vectors to safely produce beneficial biomolecules that are able to target specific endogenous molecules or specific xenobiotic pathogens [19]. The development of gene editing methods, such as clustered regularly interspaced palindromic repeat-associated proteins, is a breakthrough in the field of engineering. Utilizing such genome editing tools, probiotics are emerging and expanding their applicability to treat diseases and contribute to human health [19]. Further, synbiotics and postbiotics might also be applicable to treat AD by modulating GM.

A report that showed a high prevalence of *Helicobacter pylori* infection in patients with AD suggested that therapeutic eradication of this bacterium may improve the degenerative process in AD [20]. Another study revealed the possible neuroprotective effect of cycloserine against aluminum chloride-induced AD in rats [21]. Administration of aluminum chloride caused oxidative damage and neurodegeneration compared to the control group, and it was found that aluminum chloride decreased α-secretase activity while increasing the activities of both β-secretase and γ-secretase. On the other hand, cycloserine application improved the degree of neurodegeneration and oxidative damage caused by aluminum toxicity. It is believed that the results of this study will contribute to the synthesis of novel drugs with improved potential against neurodegeneration caused by aluminum, cognitive impairment, and medicinal development research [21]. Moreover, it has been reported that the consumption of a mixture of probiotics could affect cognitive function and some metabolic statuses in AD patients [26].

It is still unknown whether the manipulation of GM in the therapeutic application for AD can be achieved by using antibiotics or probiotics. The actions of antibiotics could be wide-ranging and even have the opposite effect, depending on the type of antibiotic used and the specific role of the microbiome in the pathogenesis of AD (Table 2). Recently, it has been reported that the long-term treatment of an established antibiotic cocktail, ABX, containing kanamycin, gentamicin, colistin, metronidazole, and vancomycin, resulted in reduced Aβ deposition only in the aggressive male APP_SWE_/PS1_L166P_ (APPPS1-21) mouse model of Aβ amyloidosis [27].

## 5. Circadian Rhythms and Microbiota

The circadian clock (circadian rhythm) is persistent even under constant conditions, with a time period of around one day as an intrinsically working clock system. The master regulators for circadian clocks are the pacemaker neurons housing serve in a hierarchical network of internal clocks, driving sleep and awake cycles and orchestrating the rhythms present in peripheral tissues [32]. With regard to the therapeutic opportunities for the medical application of this circadian knowledge, neurodegenerative diseases might be one of the promising targets that effective disease-modifying treatments are lacking [32]. The dysregulated circadian rhythms of sleep–wake cycles, melatonin levels, and Per gene expressions in patients with a wide range of neurodegenerative diseases, including AD and Parkinson’s disease, are evident. Notably, neuropathological damage is also evident in the suprachiasmatic nucleus or nuclei, a paired neuronal structure located in the anteroventral hypothalamus, and has been assumed to be a consequence of, rather than a contributor to, these disorders [4]. Many patients with neurodegenerative diseases, including those with preclinical AD, exhibit circadian clock dysfunction, which has been implicated as a potential contributor to AD pathogenesis. The circadian clock regulates immune responses in a variety of peripheral innate and adaptive immune cell types, and the circadian clock can affect the inflammatory responses of astrocytes and microglia [33]. Indeed, as Allada and Bass reported, preclinical and clinical studies have revealed a correlation between the disruption of circadian cycles and the accumulation of neurotoxic proteins and neurodegeneration [32]. They also stated that circadian control of oxidative or proteotoxic stress may play a role in neurodegeneration, and these findings point to the potential application of the regulation of circadian rhythms in new methods of treatment for neurodegenerative diseases [32].

Recently, a research group at the University of Texas Southwestern Medical Center showed that the intestinal microbiota generates diurnal rhythms in innate immunity that synchronize with time-dependent patterns of food intake to anticipate in advance the exposure of the gut to new microbes in food [34]. They showed that the rhythmic expression of antimicrobial proteins was driven by daily rhythms in the attachment of segmented filamentous bacteria, members of the mouse intestinal microbiota, to the epithelial cells, which were driven by the circadian clock generated by controlling feeding-time rhythms [34]. They also showed that, mechanistically, rhythmic segmented filamentous bacteria attachment activated an immunological circuit involving group 3 innate lymphoid cells. This circuit triggered oscillations in the expression and activation of the epithelial signal transducer and activator of transcription 3 that, in turn, produced a rhythmic expression of antimicrobial proteins and caused a variation across the day–night cycle in the resistance to *Salmonella typhimurium* [34]. Thus, the food intake rhythms of the host synchronize with GM to induce rhythmic changes in the intestinal innate immunity that can anticipate exposure to exogenous microbes, although probiotics have not been tested in this context [34]. Chronic sleep disruption in humans is related to increased susceptibility to infection. For example, people who work at night show elevated susceptibility to bacterial and viral infections compared to people who work during the day [29]. These findings suggest that altered feeding behavior could be a causative factor. The identification of mechanisms by which the circadian clock drives diurnal rhythms in the functioning of GM can illuminate its impact on host immunity [29]. Such studies could lead to the development of novel time-based therapeutic interventions for neurodegenerative diseases, including Alzheimer’s disease [34].

## 6. Perspective

Given that convincing evidence has demonstrated the roles of GM in the pathogenesis of AD, the modulation of GM may be a promising therapeutic option for AD prevention. However, most studies on gut microbiota have merely provided its relevance to AD, but the causal link between the activity of specific microorganisms and brain dysfunctions and/or AD pathology has not been fully elucidated. Therefore, it is critical to optimize different factors including combinations of microbe strains, micronutrients, and circadian rhythms, including time of treatment and physical exercise.

Physical exercise is generally recognized as beneficial to almost all aspects of human health, slowing cognitive aging and neurodegeneration [30]. The benefits of physical exercise on cognitive abilities are probably associated with increased neuronal plasticity and reduced neuroinflammation in the hippocampus. In spite of this evidence, little is known about the factors and molecular mechanisms underlying these effects. Proteomic analyses of blood plasma have revealed a concerted increase in the inhibitors of the complement cascade, including clusterin [30]. Intravenously injected clusterin bound to the brain vascular endothelial cells and reduced the expression of neuroinflammatory genes in acute brain inflammation and AD models [30]. Clinically, the degree of cognitive impairment positively correlates with higher plasma levels of clusterin [35]. Together, these findings suggest that there may be anti-inflammatory exercise factor(s) that are transferable and beneficial for the proper machinery of the brain functioning in humans who constantly engage in certain exercises [35].

Álvarez-Mercado’s research group proposed that gut microbiota plays a significant role in obesity progression [36]. They examined the regulatory mechanism of inflammatory signal pathways through regular physical activity for its application as a possible nonpharmacological-based therapy [36]. Studies that describe changes in gut microbiota have stated that regular physical activity could increase both the variance of microbiota and the ratio of the bacteria *Firmicutes* sp. versus *Bacteroidetes* sp., and both actions could suppress the progression of obesity and diminish body weight [36]. The regulation of gut microbiota by regular physical activity might be an attractive therapeutic option for improving health and controlling the progression of neurodegeneration [36]. To this end, the molecular mechanism needs to be clarified before this intervention is to be implemented according to established priorities [36].

## 7. Conclusions

A pharmacological therapy significantly effective in the treatment of Alzheimer’s disease by impeding the progression of the disease has not yet been established. Thus, the development of novel therapies effective in mitigating Alzheimer’s disease and the elucidation of their neuroprotective mechanisms are intriguing for researchers. Recent studies, including those with experimental AD models, revealed that the manipulation of gut microbiota resulted in significant mitigation of tau-related Alzheimer’s disease pathology and neurodegeneration. This implies that the modulation of gut microbiota might be a promising therapeutic option to prevent Alzheimer’s disease. The potential approach to practical therapeutic intervention for the improvement of microbiota-related diseases might be probiotics. Synbiotics and postbiotics might be also applicable to treat Alzheimer’s disease by modulating gut microbiota. These diet-based interventions are generally regarded as safe, and thus potentially more advantageous than drug-based therapies. Beyond diet, circadian control of oxidative or proteotoxic stress may play a role in neurodegeneration. Such studies could lead to the development of novel time-based therapeutic interventions for Alzheimer’s disease.

## Figures and Tables

**Table 1 life-13-01466-t001:** The effects of probiotics on phenotypes associated with AD.

Probiotic	Effect	Reference
*Bifidobacterium breve*	prevented cognitive dysfunction	[20]
*Lactobacillus acidophilus*	improved memory deficit	[23]
*Lactobacillus fermentum*	improved memory deficit	[23]
*Bifidobacterium lactis*	improved memory deficit	[23]
*Bifidobacterium longum*	improved memory deficit	[23]
*Lactobacillus plantarum*	decreased amyloid accumulation	[24]
Lactic acid bacteria and Bifidobacteria (SLAB51)	decreased amyloid accumulation	[25]
*Lactobacillus casei*	improved cognitive function	[26]
*Bifidobacterium bifidum*	improved cognitive function	[26]
Probiotic cocktail ^1^	improved cognitive function	[26]

^1^ The probiotic cocktail containing *Lactobacillus acidophilus*, *Lactobacillus casei*, *Bifidobacterium bifidum*, and *Lactobacillus fermentum* [26].

**Table 2 life-13-01466-t002:** The effects of antibiotics on the symptoms of AD.

Antibiotic	Targeted Bacterium	Effect	Reference
Amoxicillin	*Helicobacter pylori*	Cognitive status improved	[28]
Cycloserine	Gram-negative and -positive bacteria	Neurodegeneration improved	[29]
Doxycycline	Gram-negative and -positive bacteria	Neurodegeneration improved	[30]
Rifampicin	RNA synthesis blockade	Reduced amyloid-β	[31]
Vancomycin, neomycin, and pimaricin	Gram-negative and -positive bacteria	Reduced amyloid-β	[26]
ABX ^1^	Gram-negative and -positive bacteria	Neurodegeneration improved	[27]

^1^ ABX, an antibiotic cocktail containing kanamycin, gentamicin, colistin, metronidazole, and vancomycin.

## Data Availability

There are no additional data outside of this article.

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
