# Peer review of "Therapeutic Implications of Probiotics in the Gut Microbe-Modulated Neuroinflammation and Progression of Alzheimer’s Disease"

_life, 2023, doi:10.3390/life13071466_

Round 1

Reviewer 1 Report

Therapeutical approach for Alzheimer's disease is has been increasing significantly and researcher are trying various approaches; probiotics is a newer approach. 

Authors from these manuscripts tried to explain the importance of probiotics in Alzheimer's disease. However, the manuscript has few drawbacks.

1. Organization of manuscript

2. As per the title manuscript should contain info on neuroinflammation, various therapeutic strategy for Alzheimer's and Probiotics role in decreasing neuroinflammation mediated Alzheimer's disease. I am confused why the author brought oxidative stress and circadian rhythms.

3. Author should discuss or provide few mechanisms on how these probiotics are improving cognitive decline and Alzheimer's disease. 

4. Perspective section has no focus.

5. It should be better to add graphical abstract 

Author Response

> Reviewer 1

> 1. Organization of manuscript

  We thank the reviewer for the constructive comments, which have helped us significantly improve the manuscript. According to this comment, we have improved the organization by adding several sentences. Please see below for the details.

> 2. As per the title manuscript should contain info on neuroinflammation, various therapeutic strategy for Alzheimer's and Probiotics role in decreasing neuroinflammation mediated Alzheimer's disease. I am confused why the author brought oxidative stress and circadian rhythms.

The reason why we brought up a topic of oxidative stress in this context is that oxidative stress occurs and is a constant feature of the AD brain pathology. Also, the oxidative stress has a contributing role to the neuroinflammation and the pathogenesis of AD. Thus, to go into more detail on this matter, we have added sentences to the revised manuscript (lines 61-63; reference no.5).

  We developed the topic into circadian rhythms as stated in our original manuscript: “there is a correlation between the disruption of circadian cycles and the accumulation of neurotoxic proteins and neurodegeneration: circadian control of oxidative or proteotoxic stress may play a role in neurodegeneration, and these findings point to the potential application of regulation of circadian rhythms in new methods of treatment for neurodegenerative diseases” (lines 300-305 of the original manuscript). To further explain this, we have added sentences (lines 316-321 of the revised manuscript).

> 3. Author should discuss or provide few mechanisms on how these probiotics are improving cognitive decline and Alzheimer's disease. 

In accordance with the advice, we provided mechanism on how the probiotics are improving cognitive decline and Alzheimer's disease (lines 223-227).

> 4. Perspective section has no focus.

  We have modified the Perspective section accordingly.

> 5. It should be better to add graphical abstract 

  We added a graphical abstract which capture the content of the article for readers at a single glance.

Reviewer 2 Report

The authors have written a review article titled “Therapeutic implications of probiotics in the gut microbe-modulated neuroinflammation and progression of Alzheimer’s disease”. This is an interesting topic; however, the reviewers need to cite some of recent articles available in the field of AD and probiotics. Example

Intestinal Flora Balance Therapy Based on Probiotic Support Improves Cognitive Function and Symptoms in Patients with Alzheimer's Disease: A Systematic Review and Meta-analysis doi: 10.1155/2022/4806163

Efficacy of probiotics on cognition, and biomarkers of inflammation and oxidative stress in adults with Alzheimer's disease or mild cognitive impairment - a meta-analysis of randomized controlled trials DOI: 10.18632/aging.102810

Along with this, there are other suggestions:

1.      Probiotic supplementation has not always resulted in improvement in AD phenotype in preclinical or clinical studies. It is suggested that authors should add another column in table 1. Or generate another table to discuss the negative results (if any) or no effects in AD pathology.  In addition, authors have mainly focused on a single probiotic in table, there must be several other studies done using a cocktail or mixtures of bacterias as a probiotic in AD. Authors should either explain/justify why they just picked up single strain probiotic for the discussion in this review? It is suggested to add literature based on combination/cocktail of bacteria (probiotics) tested in AD.

2.      As the title of manuscript says, “Therapeutic implications of probiotics”, it is suggested to add more discussion in probiotic section of the review such as mechanism of action of probiotics in AD. Also, authors have added a section “circadian rhythms and microbiota” where a reader expects to see what probiotics have been tested in this context and what were the outcomes. If none of the literature is available, authors can write the same in the discussion that probiotics has not been tested in this context.

3.      In table. 2 authors are suggested to add other antibiotics tested in mouse models of Alzheimer disease examples:

https://pubmed.ncbi.nlm.nih.gov/31097468/

https://pubmed.ncbi.nlm.nih.gov/34572019/

4.      In conclusion section, “A recent study revealed that the manipulation of gut microbiota resulted in significant mitigation of tau-related Alzheimer’s disease pathology and neurodegeneration”, this line seems incomplete. Authors are suggested to write if this was observed in a human or animal study. And in conclusion, it should be more like a summary of work done in “Gut microbiota manipulation in AD” and not a single study based.  It is suggested to conclude if probiotics have shown beneficial effects in AD (why/how) or if it has limited effects in brain/AD pathology (why). 

Author Response

Reviewer 2

> The authors have written a review article titled “Therapeutic implications of probiotics in the gut microbe-modulated neuroinflammation and progression of Alzheimer’s disease”. This is an interesting topic; however, the reviewers need to cite some of recent articles available in the field of AD and probiotics. Example

> Intestinal Flora Balance Therapy Based on Probiotic Support Improves Cognitive Function and Symptoms in Patients with Alzheimer's Disease: A Systematic Review and Meta-analysis doi: 10.1155/2022/4806163

> Efficacy of probiotics on cognition, and biomarkers of inflammation and oxidative stress in adults with Alzheimer's disease or mild cognitive impairment - a meta-analysis of randomized controlled trials DOI: 10.18632/aging.102810

Following the comments, we have added these references as no.21 and no.22. Accordingly, we added sentences introducing these excellent meta-analysis studies to the text (lines 214-218).

> Along with this, there are other suggestions:

> 1. Probiotic supplementation has not always resulted in improvement in AD phenotype in preclinical or clinical studies. It is suggested that authors should add another column in table 1. Or generate another table to discuss the negative results (if any) or no effects in AD pathology. 

We thank the reviewer for constructive comments. Unfortunately, however, it is difficult for us to find appropriate papers to include at this stage. We would be grateful if you could let us know such articles on this matter.

> In addition, authors have mainly focused on a single probiotic in table, there must be several other studies done using a cocktail or mixtures of bacterias as a probiotic in AD. Authors should either explain/justify why they just picked up single strain probiotic for the discussion in this review? It is suggested to add literature based on combination/cocktail of bacteria (probiotics) tested in AD.

Thank you for the insightful comments. Accordingly, we have added literature based on combination/cocktail of bacteria (probiotics) tested in AD to Table 1 (reference no.26).

> 2. As the title of manuscript says, “Therapeutic implications of probiotics”, it is suggested to add more discussion in probiotic section of the review such as mechanism of action of probiotics in AD.

Following the comments, we added more discussion on the mechanism underlying the action of probiotics in Alzheimer's disease (lines 223-227).

> Also, authors have added a section “circadian rhythms and microbiota” where a reader expects to see what probiotics have been tested in this context and what were the outcomes. If none of the literature is available, authors can write the same in the discussion that probiotics has not been tested in this context.

We added a statement that probiotics has not been tested in this context (lines 341-342).

> 3. In table. 2 authors are suggested to add other antibiotics tested in mouse models of Alzheimer disease examples:

https://pubmed.ncbi.nlm.nih.gov/31097468/

https://pubmed.ncbi.nlm.nih.gov/34572019/

We added these studies to Table 2 and cited them as references no.26 and no.27.

> 4. In conclusion section, “A recent study revealed that the manipulation of gut microbiota resulted in significant mitigation of tau-related Alzheimer’s disease pathology and neurodegeneration”, this line seems incomplete. Authors are suggested to write if this was observed in a human or animal study.

Following the comments, we added a statement that it was observed in AD experimental model (lines 387-390).

> And in conclusion, it should be more like a summary of work done in “Gut microbiota manipulation in AD” and not a single study based.  It is suggested to conclude if probiotics have shown beneficial effects in AD (why/how) or if it has limited effects in brain/AD pathology (why). 

In accordance with the advice, we added a sentence to the Conclusion section (lines 394-396). We wish to thank the reviewer again for the valuable comments.

Round 2

Reviewer 1 Report

Author has addressed reviewer's comments eloquently.